# Quantum Image Encryption Based on Quantum DNA Codec and Pixel-Level Scrambling

**DOI:** 10.3390/e25060865

**Published:** 2023-05-29

**Authors:** Jie Gao, Yinuo Wang, Zhaoyang Song, Shumei Wang

**Affiliations:** 1School of Information and Control Engineering, Qingdao University of Technology, Qingdao 266520, China; 2School of Science, Qingdao University of Technology, Qingdao 266520, China

**Keywords:** quantum information, DNA coding, quantum image encryption, quantum image scrambling

## Abstract

In order to increase the security and robustness of quantum images, this study combined the quantum DNA codec with quantum Hilbert scrambling to offer an enhanced quantum image encryption technique. Initially, to accomplish pixel-level diffusion and create enough key space for the picture, a quantum DNA codec was created to encode and decode the pixel color information of the quantum image using its special biological properties. Second, we used quantum Hilbert scrambling to muddle the image position data in order to double the encryption effect. In order to enhance the encryption effect, the altered picture was then employed as a key matrix in a quantum XOR operation with the original image. The inverse transformation of the encryption procedure may be used to decrypt the picture since all the quantum operations employed in this research are reversible. The two-dimensional optical image encryption technique presented in this study may significantly strengthen the anti-attack of quantum picture, according to experimental simulation and result analysis. The correlation chart demonstrates that the average information entropy of the RGB three channels is more than 7.999, the average NPCR and UACI are respectively 99.61% and 33.42%, and the peak value of the ciphertext picture histogram is uniform. It offers more security and robustness than earlier algorithms and can withstand statistical analysis and differential assaults.

## 1. Introduction

With the development of internet and communication technology, image has become the most widely used information transmission medium. Compared with text information, images contain more information. As a result, researchers suggest quantum image processing by extending the digital picture to the quantum computing framework [1,2].

Quantum image processing (QIP) is committed to using quantum computing technology to capture, restoration, and other classical image operations. its exponential storage capacity and parallelism give this technology a strong advantage in implementing operations requiring high real-time operations such as image retrieval and processing.

The special behavior of quantum particles is regarded as the rules of quantum physics and the tool of mathematical logic. In 1992, Lucien Hardy proposed Hardy’s paradox while proving Bell’s theorem [3], and continued to prove the nonlocality of two particles in 1993 [4]. Moreover, Shor [5] and Grover [6] created quantum algorithms using the same quantum computer property for integer factoring and database searching, respectively. These algorithms perform better than their traditional versions in terms of running time. References [5,6], which laid the foundation for diverse applications of quantum computing in the information sciences, served as an inspiration for a great number of researchers [7,8,9,10,11,12,13,14,15,16].

Quantum image encryption can be “unconditionally secure” based on the Heisenberg uncertainty principle because of quantum features such as quantum entanglement, coherence, parallelism, and superposition. Quantum picture encryption employs the “No-Cloning Theorem”, derived from the Heisenberg uncertainty principle, to encrypt image data, whereas conventional encryption typically restricts the timeliness of decryption operations. That is, because the basis of replication is measurement, and because measurement often modifies the quantum state, it is impossible to accomplish the process of accurate duplication of any unknown quantum state in quantum mechanics.

To establish image protection in the sphere of digital pictures [17], the genuine and meaningful images are often transformed into meaningless forms. Today’s latest research hot topic is the quantum picture encryption technique created by fusing together quantum computing and digital imaging [18,19,20]. There are several quantum image representation techniques now being used [21], including FRQI [22], NEQR [23], MCQI [24], NASS [25,26], QUALPI [27], and others. Image information encryption has caught the attention of academics working in the area of quantum information processing. Recently, several quantum image encryption methods have been created, including a quantum image encryption scheme based on quantum image decomposition [28], an iterative extended Arnold transform-based quantum image encryption method, and a quantum image cyclic shift operation-based quantum image encryption strategy [29].

DNA coding has attracted wide attention because of its advantages such as large storage capacity and strong parallel processing ability. Compared with the traditional cryptography based on mathematical problems, DNA cryptography combines the two fields of mathematics and biology, which greatly enhances the security and robustness of DNA cryptography. In 1994, Adleman carried out the world’s first DNA computing experiment [30] and published related results in the journal Science. This result revealed that DNA molecules have computing power in addition to their stable genetic properties, and have since opened up a new information age [31,32,33]. At present, DNA coding is also gradually emerging in the field of encryption [34,35,36,37]. Scholars have proposed a classical image encryption algorithm that combines DNA coding technology with quantum walking [38].

In this research, a DNA coding technique and picture Hilbert scrambling were combined to develop a quantum image encryption scheme. The encryption technique uses Hilbert quantum image scrambling and quantum picture DNA coding and decoding. By closely integrating the two technologies, the goal of enhancing picture security may be achieved by more effectively reducing the high connection between neighboring pixels. We also developed the quantum DNA codec’s implementation circuit.

The rest of this article is structured as follows: Section 2 introduces the background. Section 3 shows the quantum circuit model. In Section 4, the flow of encryption and decryption algorithm is shown in detail. Section 5 introduces the theoretical analysis and experimental simulation. Finally, Section 6 draws a conclusion.

## 2. Related Work

### 2.1. Quantum Color Image Representation

NCQI is a quantum color digital image representation method proposed in 2017 [39]. We briefly reviewed the NCQI quantum representation model so as to introduce the quantum image encryption algorithm proposed in this paper.

The NCQI model of a $2^{n}\times 2^{n}$ image |ψ〉 can be mathematically expressed as follows:(1)$$\begin{array}{c} \begin{array}{c} |\psi \rangle =\frac{1}{2^{n}}\sum_{y=0}^{2^{n}-1}\sum_{x=0}^{2^{n}-1}{|c}_{\left(y,x\right)}\rangle ⊗|yx\rangle , \end{array} \end{array}$$
where $|c_{\left(y,x\right)}\rangle$ represents the color value of the pixel, which can be encoded by binary sequence $R_{q-1}\cdots R_{0}G_{q-1}\cdots G_{0}B_{q-1}\cdots B_{0}$.

Every pixel contained in a color channel, which has a range of $\left[0,2^{q}-1\right]$, is represented by three components: the horizontal position *X*, the vertical position *Y*, and the color information $c_{\left(y,x\right)}$. The *R*, *G*, and *B* range $\left[0,2^{q}-1\right]$ of each channel is utilized to store picture data in an NCQI state of a color image using the 2*n* + 3*q* qubits.

Figure 1 is an example of a 4-by-4-color picture with the three channels, *R*, *G*, and *B*, with the range size $\left[0,2^{8}-1\right]$, *n* = 1, and *q* = 8. The equation in Figure 1 states that the whole NCQI is kept in a state of normalized quantum superposition, with each base standing in for a single pixel.

### 2.2. DNA Coding Method and Operation

Adenine (A), thymine (T), cytosine (C), and guanine (G) are the four nucleotides that make up the molecular structure of deoxyribonucleic acid (DNA), which is based on the biological model (G). The DNA pairing rule states that A and T pair and C and G pair. Similarly, in the binary complementary calculation, 1 and 0 are complementary, and eight coding schemes which accord with the rules of a biological model were obtained by encoding each nucleic acid base with 2-bit binary number respectively, as shown in Table 1. Each RGB three-channel pixel in a color image is represented as a 24-bit binary sequence as part of the encryption procedure, where each color channel’s 8-bit binary sequence may be represented by four bases. For instance, scheme 1 will produce CACT if an image pixel’s R-channel gray value is 71, which is represented by the binary sequence 01000111.

### 2.3. Quantum Hilbert Scrambling

With the development of quantum image processing, many image scrambling methods have emerged [40,41]. In this work, quantum Hilbert image scrambling [42] was used.

An original image of size $2^{n}\times 2^{n}$ can be regarded as a matrix. We call this matrix the starting matrix (or original matrix) $S_{n}$ and used 1 to $2^{2n}$ to encode all pixels,
(2)$$\begin{array}{c} S_{n}=\left(\begin{array}{ccccc} 1 & 2 & 3 & \cdots & 2^{n}\\ 2^{n}+1 & 2^{n}+2 & 2^{n}+3 & \cdots & 2^{n+1}\\ \vdots & \vdots & \vdots & \vdots & \vdots \\ 2^{2n-1}+1 & 2^{2n-1}+2 & 2^{2n-1}+3 & \cdots & 2^{2n}. \end{array}\right) \end{array}$$

The arrangement of $S_{n}$ is $H_{n}$. For example, $H_{0}=\left(1\right)$, $H_{1}=\left(\begin{array}{cc} 1 & 2\\ 4 & 3 \end{array}\right)$, $H_{2}=\left(\begin{array}{cccc} 1 & 2 & 15 & 16\\ 4 & 3 & 14 & 13\\ 5 & 8 & 9 & 12\\ 6 & 7 & 10 & 11 \end{array}\right)$, where $H_{n}\left(i,j\right)$ represents the pixel at position (i,j) of the starting matrix $S_{n}$. Hilbert curves (see Figure 2) and scrambled images (see Figure 3) can be obtained along the $H_{n}$.

This paper adopted the improved Hilbert scrambling recursive generation algorithm in [42]. If *A* is a matrix, then $A^{T}$ represents its transposition, $A^{ud}$ its upper and lower direction reversed, $A^{ld}$ its left and right inversion, and $A^{\mathrm{pp}}$ its centre rotation matrix. For a quantum computer to implement Hilbert image scrambling,
(3)$$\begin{array}{c} H_{n+1}=\left\{\begin{array}{c} \left(\begin{array}{cc} H_{n} & {\left(H_{n}+4^{n}E_{n}\right)}^{T}\\ {\left(H_{n}+3\times 4^{n}E_{n}\right)}^{\mathrm{pp}} & {\left(H_{n}+2\times 4^{n}E_{n}\right)}^{T} \end{array}\right),n\;\mathrm{is}\;\mathrm{even}\;\\ \left(\begin{array}{cc} H_{n} & {\left(H_{n}+3\times 4^{n}E_{n}\right)}^{\mathrm{pp}}\\ {\left(H_{n}+4^{n}E_{n}\right)}^{T} & {\left(H_{n}+2\times 4^{n}E_{n}\right)}^{T} \end{array}\right),n\;\mathrm{is}\;\mathrm{odd}\; \end{array}\right., \end{array}$$
where *n* is a positive integer and the initial matrix is $H_{1}=\left(\begin{array}{cc} 1 & 2\\ 4 & 3 \end{array}\right)$, $E_{n}=\left(\begin{array}{cccc} 1 & 1 & \cdots & 1\\ 1 & 1 & \cdots & 1\\ \vdots & \vdots & \ddots & \vdots \\ 1 & 1 & \cdots & 1 \end{array}\right)$.

According to reference [42], initialization, and even and odd basic circuits, are also integrated circuits, and the process is described in Figure 4. The three parts that make up the three basic circuits are called three circuit modules.

### 2.4. Quantum XOR

According to reference [43], it needs to be divided into $2^{2n}$ sub-operations $Y_{y,x}$ in order to implement the XOR operation on each pixel value of the quantum image. Use *Y* to represent a matrix of the same size as the image in the sub-operation:(4)$$\begin{array}{c} Y=\left(\begin{array}{ccc} y_{0,1} & \cdots & y_{0,2^{n}-1}\\ \vdots & \ddots & \vdots \\ y_{2^{n}-1,0} & \cdots & y_{2^{n}-1,2^{n}-1} \end{array}\right), \end{array}$$
where $y_{y,x}=m_{yx}^{0}m_{yx}^{1}m_{yx}^{2}m_{yx}^{3},\cdots ,m_{yx}^{23}$, $m_{yx}^{i}\in 0,1$, and the quantum gate operation sequence *F* are generated according to $y_{i,j}$:(5)$$\begin{array}{c} F=\left(\begin{array}{ccc} b_{0,1} & \cdots & b_{0,2^{n}-1}\\ \vdots & \ddots & \vdots \\ b_{2^{n}-1,0} & \cdots & b_{2^{n}-1,2^{n}-1} \end{array}\right), \end{array}$$
where $F_{yx}=V_{yx}^{0}V_{yx}^{1}\ldots V_{yx}^{23}$, $V_{yx}^{i}=\left\{\begin{array}{c} X,m_{yx}^{i}=1\\ I,m_{yx}^{i}=0 \end{array}\right.$ represents the realization of the *X*-gate transformation or *I*-gate transformation of $C_{yx}^{i}$, respectively:(6)$$\begin{array}{c} G_{X}=\left(\begin{array}{cc} 0 & 1\\ 1 & 0 \end{array}\right),G_{I}=\left(\begin{array}{cc} 1 & 0\\ 0 & 1. \end{array}\right) \end{array}$$

When *F* is applied to the entire image, there are:(7)$$\begin{array}{c} \begin{array}{cc} & F|I\rangle =\prod_{x=0}^{2^{n}-1}\prod_{y=0}^{2^{n}-1}F_{yx}|I\rangle \\ & =\frac{1}{2^{n}}\sum_{x=0}^{2^{n}-1}\sum_{y=0}^{2^{n}-1}{⊗}_{i=0}^{23}|C_{yx}^{i}⊗m_{yx}^{i}\rangle |yx\rangle \\ & =\frac{1}{2^{n}}\sum_{x=0}^{2^{n}-1}\sum_{y=0}^{2^{n}-1}|f\left(y,x\right)\rangle |yx\rangle , \end{array} \end{array}$$
where |f(Y,X)〉 represents the new pixel value after pixel scrambling, and $|C_{YX}\rangle$ is the pixel sequence.

## 3. Quantum Circuit Design

The design of the DNA codec simulator’s quantum circuit, which is a crucial component of our quantum picture encryption technique, is presented in this section.

### 3.1. Quantum DNA Codec Simulator

Based on the classical DNA coding technology and the quantum image representation of NCQI, a DNA codec simulation circuit for quantum images was designed.

In our proposed encryption algorithm, quantum DNA coding and decoding technology were used to encrypt pixel color information. NCQI representation can directly transform the color value information of a color image with three-channel color values in the range of $\left[0,2^{q}-1\right]$ of $2^{n}\times 2^{n}$ into a binary sequence of 3*q* bits, so we took every two lines as a combination to reflect the concept of bases in biology.

We encapsulated the whole quantum DNA codec into a black box. We only needed to input the binary sequence of the image into the black box and enter the sequence number of the coding and decoding scheme to complete the DNA codec operation of the quantum image. $D_{i,j}$ was used to represent the quantum DNA codec, where *i* is the coding scheme sequence number and *j* is the decoding scheme sequence number. As shown in Figure 5, six lines were added to reflect the sequence number of the codec scheme, and $|\psi_{0}\rangle$ and $|\psi_{1}\rangle$ were input lines as binary sequences. Three quantum lines $b_{0}$,$b_{1}$,$b_{2}$ can represent the numerical value of the decoding scheme sequence number. When designing quantum circuits, we used two auxiliary circuits. While reducing a lot of time complexity, we only added a little space complexity. Therefore, it can effectively reduce the circuit complexity and improve the operation efficiency. If we decode it with option 6, the circuit module of the scheme will be run directly through the three quantum lines $b_{0}$,$b_{1}$,$b_{2}$. As a result, only the sequence number of the decoding scheme can be input to run the circuit to realize automatic decoding, and there is no need to select the circuit. Figure 6 shows the quantum circuits of seven decoding schemes encoded in Rule 1.

In this paper, we designed the quantum circuits of seven kinds of decoders with scheme one as the coding scheme, and showed the process of transforming the same binary sequence from scheme one to the other seven schemes. As shown in Table 1, the sequence was first quantum DNA encoded according to scheme 1. If the second scheme is used for decoding, it is necessary to reverse the two lines when the high qubits are different from the low qubits, that is, to realize the interchange between C and G. If the third scheme is used for decoding, it is necessary to reverse all the low qubits, that is, to realize the interchange between A and C and between T and G. If we use scheme 4 to decode, we need to reverse the high qubit and the low qubit at the same time and, if the high qubit is different from the low qubit, we need to flip the low qubit. If we use scheme 5 to decode, contrary to scheme 4, we need to flip high qubits when high qubits are different from low qubits, and if high qubits and low qubits flip low qubits at the same time. If decoding is carried out in scheme 6, each set of high qubits needs to be flipped. If we use scheme 7 to decode, when the high qubit and the low qubit are all flipped, the interchange between An and T will be realized. If we use scheme 8 to decode, it is necessary to reverse all the high and low qubits, that is, to realize the interchange between A and T and between C and G.

### 3.2. Hilbert Image Scrambling Quantum Circuit

The Hilbert scrambling operation of quantum image was divided into three steps: quantum image partition and Hilbert image scrambling parity operation, in which the parity operation is carried out alternately. The composition of these three basic circuits is described below. In this section, *k* is an integer and 0≤k≤n−1.

#### 3.2.1. Initialization Module

The initialization quantum module is beneficial to the segmentation of the quantum image and the formation of $H_{n}$, and the partition module PARTITION (K) plays a major role. The PARTITION (*k*) module can divide the input image of size $2^{n}\times 2^{n}$ into $2^{n-k-1}\times 2^{n-k-1}$ blocks of size $2^{k+1}\times 2^{k+1}$; the initialization module quantum circuit is shown in Figure 7a:

#### 3.2.2. Odd(k) Module

The main function of the Odd(k) module is to encrypt the pixel position information, where *k* is *odd* and 1≤k≤n−1. The main role is the *odd* module O(k) in the Odd(k) module. Figure 7b shows the complete quantum circuit of the O(k) module.

#### 3.2.3. Even (k) Module

As with the function of the Odd(k) module, the function of the Even (k) module is to transform the pixel position, where *k* is even and 2≤k≤n−1. Figure 7c shows the complete Even (k) quantum circuit.

## 4. Encryption and Decryption of Quantum Images

Quantum image diffusion and scrambling are the two key components of this paper’s encryption phase. At the diffusion step, the picture is made confusing by using DNA coding and various decoding techniques, and the original image is quantum XOR coded. The approach employs iterative Hilbert scrambling during the scrambling stage to encrypt the image’s pixel location data.

### 4.1. Encryption Process

Using quantum Hilbert scrambling and DNA coding technology, we designed the following quantum image encryption method. The original input quantum image size is $2^{n}\times 2^{n}$ and the image representation is NCQI. The encryption flow chart is shown below. Figure 8 shows the encryption process of Rule 1.

Step 1: The pixel matrix of the original image is divided into three RGB channels, and the NCQI representation model is loaded as a quantum image.
(8)$$\begin{array}{c} \begin{array}{c} |\psi_{1}\rangle =\frac{1}{2^{n}}\sum_{y=0}^{2^{n}-1}\sum_{x=0}^{2^{n}-1}|c\left(y,x\right)\rangle ⊗|yx\rangle . \end{array} \end{array}$$

Step 2: The quantum color image is encoded and decoded through the quantum image DNA codec.

In the NCQI representation, the RGB color information will be input into the circuit in binary form, so this paper used rule 1 in Table 1 to encode the binary sequence, and then decodes the sequence according to rule 6, that is, the quantum image is input to the quantum DNA codec, the $D_{1,6}$ operation is performed, and the output is $|\psi_{2}\rangle$.
(9)$$\begin{array}{c} \begin{array}{c} |\psi_{2}\rangle =D_{1,6}|\psi_{1}\rangle . \end{array} \end{array}$$

Step 3: Perform Hilbert quantum image scrambling with $|\psi_{2}\rangle$ iteration.

$|\psi_{2}\rangle$ has $2^{n}\times 2^{n}=2^{2n}$ pixels, and if the original image pixel order is “$1,2,3,4,\ldots ,2^{2n}$”, the partition module PARTITION (0) will separate the picture into 2×2 sub-images, that is, $\left(\begin{array}{cc} a & a+1\\ a+2 & a+3 \end{array}\right)$. The last two pixels of each sub-image are switched by the C-NOT gate, which will separate the picture into 2×2 sub-images.

The partition module PARTITION (1) divides the image into sub-images of 4×4 and so on, and operates in sequence O(1),E(2),O(3),E(4),…,O(n−1)/E(n−1) until it is executed to PARTITION (*n* − 2).

Finally, the scrambled sub-image is restored to the original image size $2^{n}\times 2^{n}$ and named $|\psi_{3}\rangle$.
(10)$$\begin{array}{c} \begin{array}{c} |\psi_{3}\rangle =Q_{2^{n}}|\psi_{2}\rangle , \end{array} \end{array}$$
where $Q_{2^{n}}$ represents performing Hilbert quantum image scrambling on an image of size $2^{n}\times 2^{n}$.

Step 4: Between the original picture and the scrambled image, use quantum XOR coding.

Generate matrix $Y_{YX}$ from the pixel color value of image $|\psi_{1}\rangle$ and convert each element into an octet binary,
(11)$$\begin{array}{c} Y_{YX}=\left(\begin{array}{ccc} y_{0,0} & \cdots & y_{0,2^{n}-1}\\ \vdots & \ddots & \vdots \\ y_{2^{n}-1,0} & \cdots & y_{2^{n}-1,2^{n}-1} \end{array}\right), \end{array}$$
where $y_{yx}=m_{yx}^{0}m_{yx}^{1}m_{yx}^{2}\ldots m_{yx}^{23},m_{yx}^{i}\in 0,1$. According to matrix $Y_{YX}$, the quantum XOR operation matrix *F* is generated, which is the same as $y_{yx}$,$b_{yx}=V_{yx}^{0}V_{yx}^{1}V_{yx}^{2}\ldots V_{yx}^{23}$, where $V_{yx}^{0}∼V_{yx}^{7}$ controls the *R* channel in the quantum circuit, $V_{yx}^{8}∼V_{yx}^{15}$ controls the *G* channel, and $V_{yx}^{16}∼V_{yx}^{23}$ controls the *B* channel, $V_{yx}^{i}=\left\{\begin{array}{c} X,m_{yx}^{i}=1\\ I,m_{yx}^{i}=0 \end{array}\right.$ obtain image $|\psi_{4}\rangle$.

Step 5: Decode the image with different rules through the quantum image DNA codec simulator.

$|\psi_{4}\rangle$ is encoded and decoded by quantum DNA codec, perform the $D_{1,7}$ operation to get $|\psi_{5}\rangle$, and the encryption is completed.
(12)$$\begin{array}{c} |\psi_{5}\rangle =D_{1,7}|\psi_{4}\rangle . \end{array}$$

### 4.2. Decryption Process

The reversibility of quantum circuits serves as the foundation for the quantum image decryption technique developed in this research. As a whole, the procedure is as follows:

Step 1: On the encrypted picture, peform the inverse quantum DNA coding and decoding procedure.

The quantum circuit module of the DNA encoder and the quantum DNA codec $D_{7,1}$ are both used to decrypt the encrypted picture to produce the result $|\psi_{4}\rangle$.

Step 2: Inverse quantum XOR coding.
(13)$$\begin{array}{c} |\psi_{3}\rangle =X_{5}^{-1}|\psi_{4}\rangle . \end{array}$$

Step 3: The quantum image is iterated to perform Hilbert inverse scrambling.

Because the biggest difference between the quantum circuit and the classical circuit is that the quantum circuit is reversible, and there is no information loss in this process, the Hilbert image scrambling quantum circuit in this paper is reversible. We can input the scrambled image $|\psi_{3}\rangle$ and obtain the reconstructed image $|\psi_{2}\rangle$ using quantum Hilbert inverse scrambling.
(14)$$\begin{array}{c} |\psi_{2}\rangle =Q_{2^{n}}^{-1}|\psi_{3}\rangle . \end{array}$$

Step 4: Pass $|\psi_{2}\rangle$ through quantum DNA codec $D_{6,1}$ to obtain the original image $|\psi_{1}\rangle$.
(15)$$\begin{array}{c} |\psi_{1}\rangle =D_{6,1}|\psi_{2}\rangle . \end{array}$$

## 5. Safety Analysis

We performed simulation experiments in MATLAB and Python using classical computers since there are no quantum computers available. We did not take into account the impact of decoherence and inaccuracy in the quantum version while processing numerical data. In order to examine the encrypted data in this part, three color pictures of pineapples, roses, and plants with pixel sizes of 512×512 were utilized as the original image. The following summarizes the encryption and decryption simulation findings. Figure 9 shows the comparison of the results before and after the application of this algorithm.

### 5.1. Histogram Analysis

One of the key indications needed to assess the security of encryption techniques is the histogram, which may considerably reflect the intensity distribution of picture pixels. The encrypted histograms of the three photos are shown in Figure 10, Figure 11 and Figure 12.

In contrast to the non-uniform peak distribution of the plaintext histogram, which is seen in the image, the peak value of the histogram encrypted by this approach becomes uniform. As a result, the attacker is unable to obtain the picture data by studying the ciphertext image’s histogram.

### 5.2. Correlation Analysis of Adjacent Pixels

A crucial metric for determining the efficacy of the encryption technique is the correlation between neighboring pixels. As there is a significant connection between neighboring pixels in the original picture, a good encryption technique should minimize this correlation to zero. In this study, we utilized this coefficient to compare the correlation between neighboring pixels both before and after the method was applied
(16)$$\begin{array}{c} r=\frac{\mathrm{cov}(x,y)}{\sqrt{D(x)D(y)}}, \end{array}$$
where *A* and *B* represent the values of adjacent pixels, cov(A,B) is the covariance of *A* and *B*, and $\sqrt{D(A)}$ and $\sqrt{D(B)}$ are the variances of *A* and *B*. In this section, the pixel correlation between the original image and the ciphertext image was analyzed horizontally, vertically, and diagonally. The results are shown in Figure 13 and Figure 14, and the specific data are reflected in Table 2 and Table 3, where C-Image represents the ciphertext image.

The suggested encryption technique clearly creates a sizable correlation gap between the ciphertext picture and the original image based on the data shown in the chart, demonstrating the algorithm’s effectiveness.

### 5.3. Key Sensitivity Analysis

Two words that are often used to characterize the quantity of pixels and the average intensity of change between the original picture and the ciphertext image are NPCR and UACI. In accordance with the associated ideal value, the key sensitivity of the NPCR = 99.6094%, UACI = 33.4635% algorithm should be as high as possible; the more closely the numerical value resembles the ideal value, the stronger the security of the encryption technique should be. The data for this method’s NPCR and UACI are shown in Table 4. Table 5 compares our work numerically to the NPCR and UACI algorithms that have been proposed in different papers. This indicates very clearly how much more efficiently the technique used in this research can guarantee picture confidentiality.

### 5.4. Information Entropy

We often use information entropy to evaluate the unpredictability of the distribution of ciphertext pictures. The ciphertext image’s pixels may be distributed evenly via a decent picture encryption method, making the image more resistant to outside attacks. The perfect information entropy is eight. The image encryption effect is better and the value is more closely aligned with the ideal value as the pixel distribution becomes more uniform. The information entropy of our recommended approach is shown in Table 6. The following table provides ample proof of the algorithm’s strong security and robustness by showing that the average information entropy of RGB’s three channels may reach 7.999.

### 5.5. Key Space

The modified picture serves as the key matrix in the encryption procedure described in this research. The color picture is 512×512 in size, making its key space $2^{512\times 512\times 24}$ pixels, which is sufficient to stave against brute force assaults.

### 5.6. Scheme Reversibility Verification

Indicators used often to assess picture quality in the field of image processing include peak signal-to-noise ratio (PSNR) and structural similarity (SSIM).

#### 5.6.1. Peak Signal-to-Noise Ratio

To evaluate the image’s decryption quality, we employed PSNR. The floating-point value that PSNR returns will range from 30 to 50 if the two input photos are comparable; the higher the number, the greater the similarity. The PSNR values of plaintext pictures and encrypted images with a size of 512 to 512 are larger than 30 dB, according to simulation findings, and the average value is 43.4590. Table 7 displays the specific data. This demonstrates the algorithm’s strong ability to aid in rebuilding.

#### 5.6.2. Structural Similarity

The SSIM value ranges from 0 to 1. The value of SSIM increases with the degree of similarity between the two photos. The picture acquired using the image decryption approach suggested in this work was compared with the original image. Table 7 displays the specific data. The average value of SSIM determined by the simulation results is 0.980358, which shows that the technology has an excellent decryption and recovery effect.

## 6. Conclusions

Quantum image processing is committed to the use of quantum computing technology to capture, restoration, and other classical image operations. Because of its exponential storage capacity and parallelism, this technology has a strong advantage in realizing real-time operations such as image retrieval and processing. In this paper, the circuit of a quantum DNA codec was designed, and the image information was encrypted by using the biological characteristics of DNA and the physical properties of quantum mechanics. At the end of this article, the combination of DNA technology and quantum image encryption was studied and verified. According to the simulation, average NPCR = 99.6094%, average NPCR = 33.4244%, the average information entropy of RGB three channels is more than 7.999, and the average value of SSIM determined by the simulation results is 0.980358. These results unmistakably demonstrate the viability and effectiveness of the quantum picture encryption system presented in this research, which is based on DNA codec and Hilbert scrambling.

In this encryption scheme, a quantum DNA codec was designed to enable the biological field to participate in the quantum image encryption process. It is hoped that it can play a greater role in the later research. In the follow-up work, we hope to combine quantum random walk with DNA technology to realize the integration of physics and biology again. This will be the focus of our next paper.

## Figures and Tables

**Figure 1 entropy-25-00865-f001:**
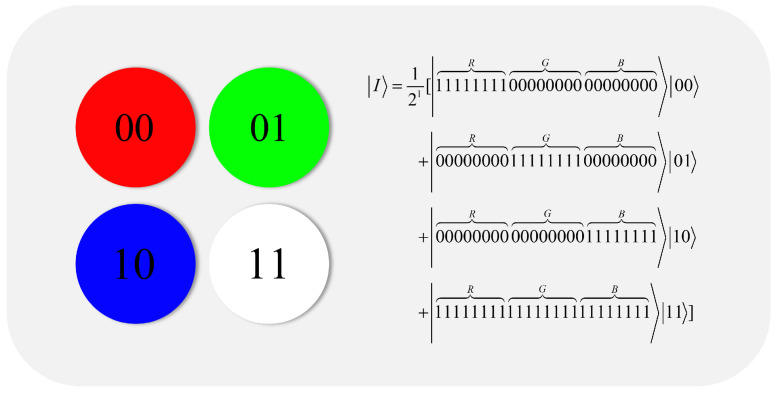
A color image and its quantum representation of NCQI.

**Figure 2 entropy-25-00865-f002:**
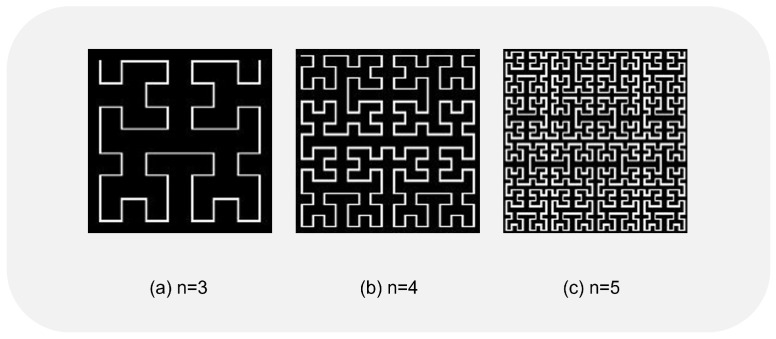
Hilbert curve.

**Figure 3 entropy-25-00865-f003:**
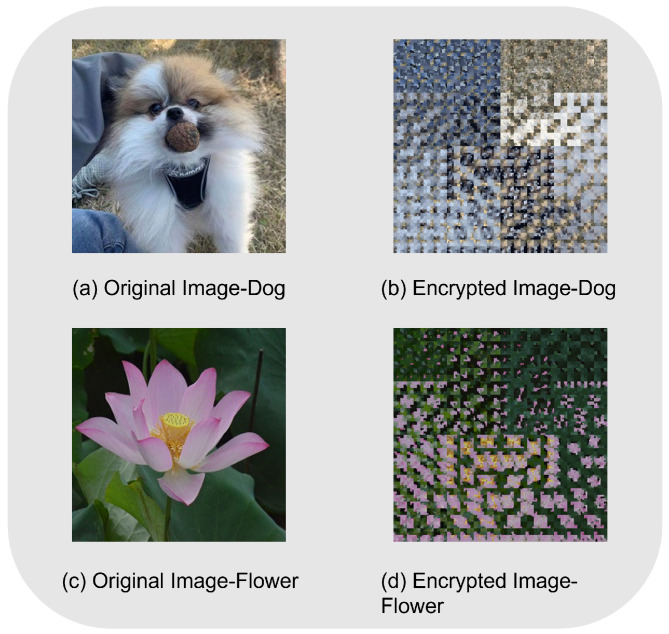
Results of performing a single Hilbert image scrambling.

**Figure 4 entropy-25-00865-f004:**
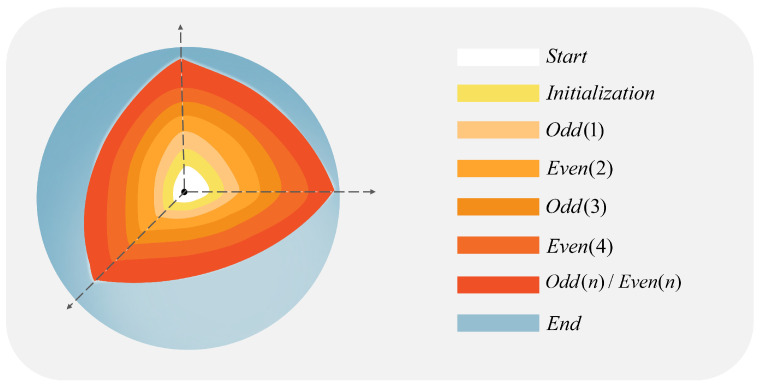
Flow chart of recursive generation algorithm.

**Figure 5 entropy-25-00865-f005:**
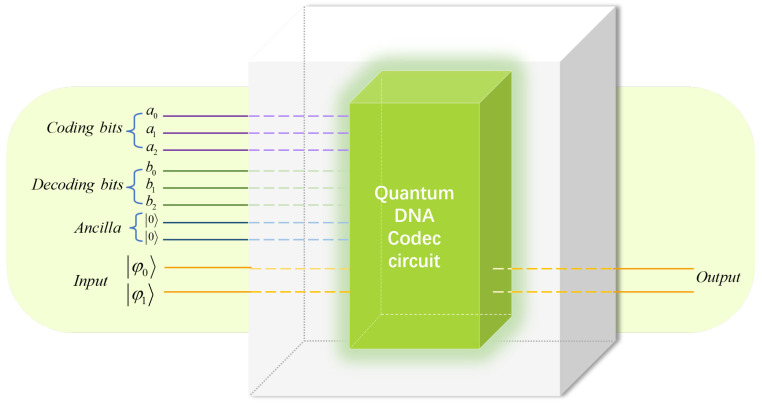
QuantumDNA codec analog circuit.

**Figure 6 entropy-25-00865-f006:**
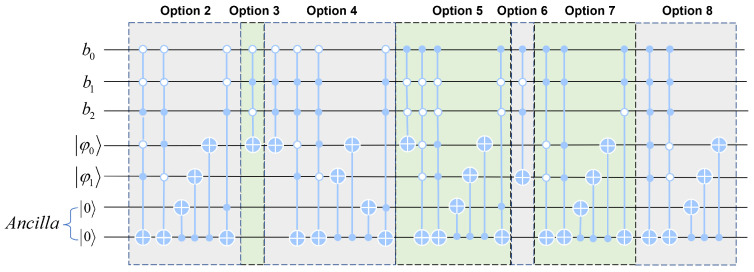
Seven kinds of quantum DNA codec simulators encoded by Rule 1.

**Figure 7 entropy-25-00865-f007:**
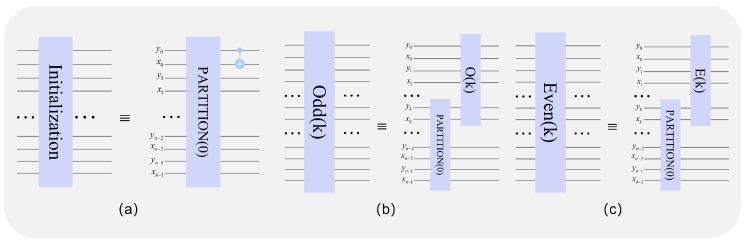
Initializing the quantum circuit. (**a**) implement the block function (**b**) implement the scrambling function when *k* is *odd* (**c**) realize the scrambling function when k is even.

**Figure 8 entropy-25-00865-f008:**
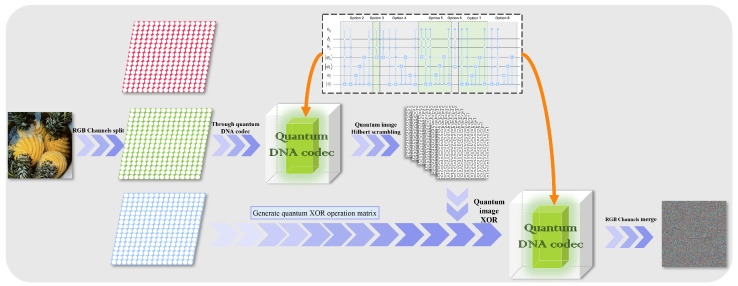
Algorithm flow chart applied to rule one.

**Figure 9 entropy-25-00865-f009:**
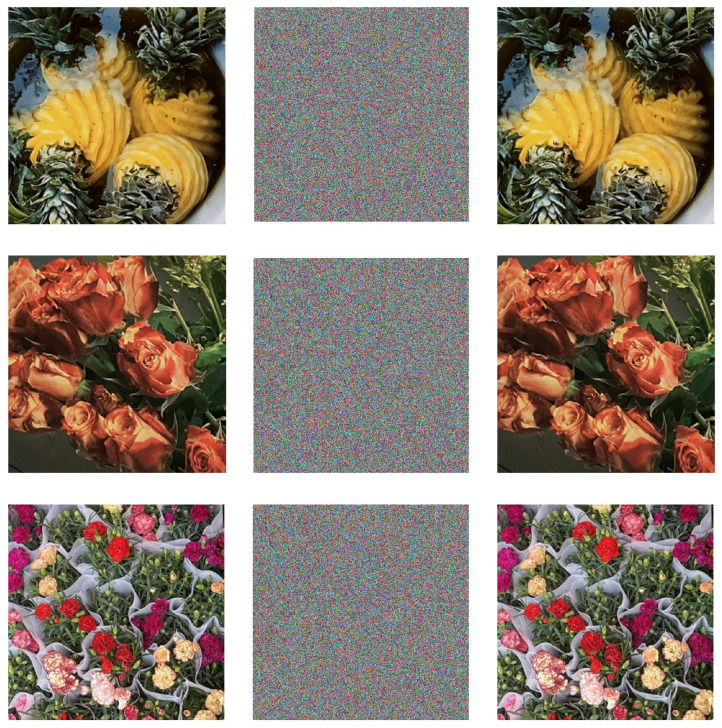
Effect of encryption and decryption.

**Figure 10 entropy-25-00865-f010:**
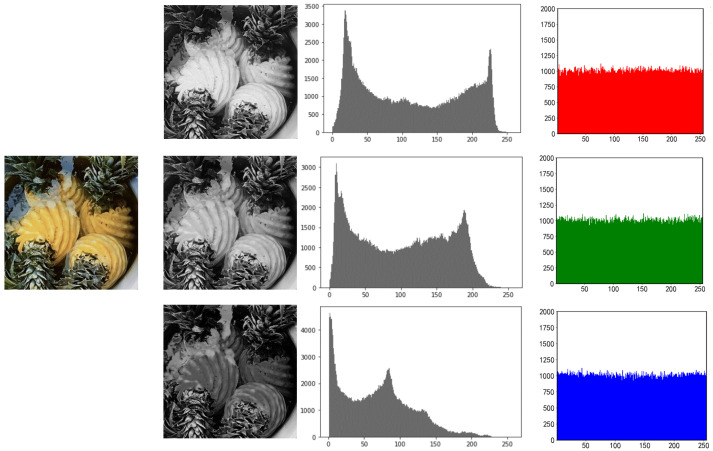
RGB three-channel histogram of pineapple before and after encryption.

**Figure 11 entropy-25-00865-f011:**
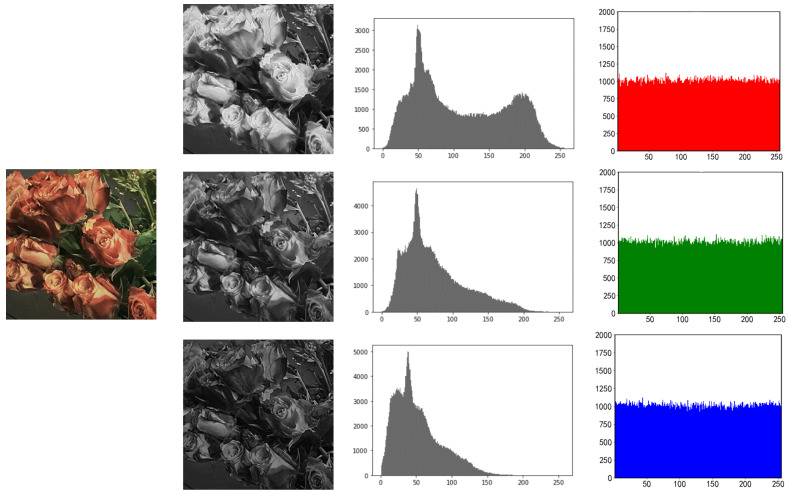
RGB three-channel histogram of rose before and after encryption.

**Figure 12 entropy-25-00865-f012:**
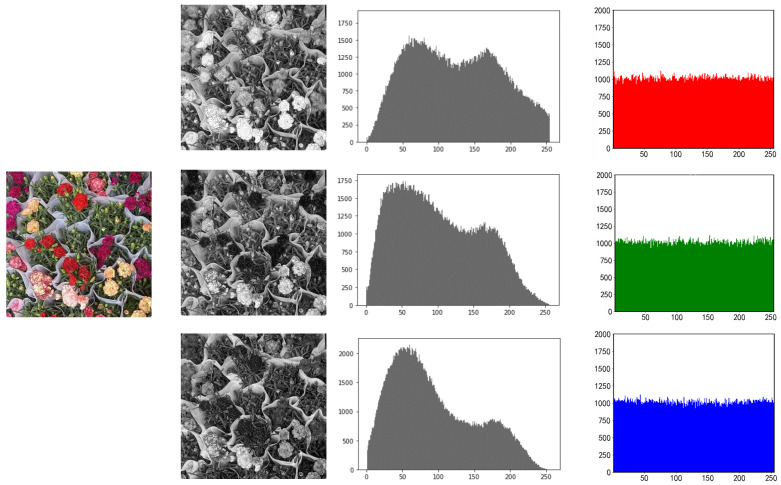
RGB three-channel histogram of plants before and after encryption.

**Figure 13 entropy-25-00865-f013:**
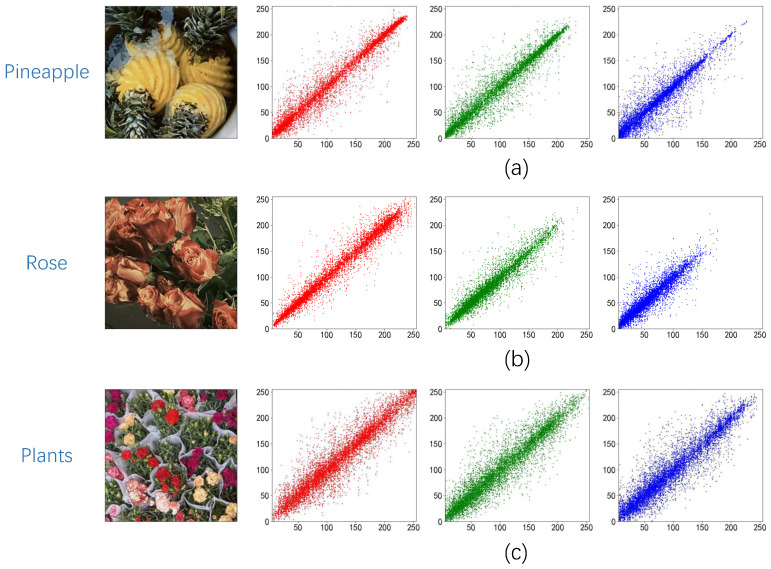
(**a**) Pineapple correlation analysis; (**b**) Rose correlation analysis; (**c**) Plants correlation analysis.

**Figure 14 entropy-25-00865-f014:**
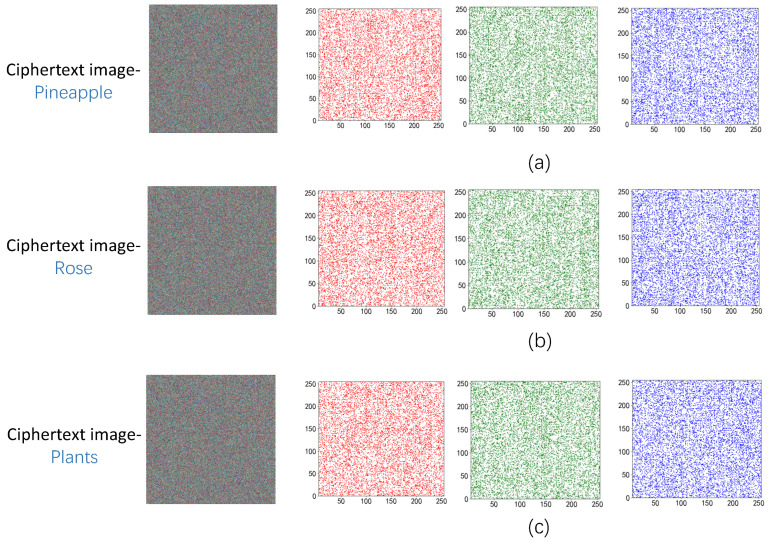
(**a**) Correlation analysis after pineapple encryption; (**b**) Correlation analysis after Rose encryption; (**c**) Correlation analysis after plants encryption.

**Table 1 entropy-25-00865-t001:** DNA coding rules.

	1	2	3	4	5	6	7	8
00	A	A	C	C	G	G	T	T
01	C	G	A	T	A	T	C	G
10	G	C	T	A	T	A	G	C
11	T	T	G	G	C	C	A	A

**Table 2 entropy-25-00865-t002:** Correlation analysis value of original image.

Image	Channel	Horizontal	Vertical	Diagonal
	R	0.9849	0.9813	0.9833
Pineapple	G	0.9753	0.9763	0.9588
	B	0.9597	0.9550	0.9251
	R	0.9835	0.9844	0.9753
Rose	G	0.9651	0.9643	0.9466
	B	0.9461	0.9446	0.9115
	R	0.9539	0.9583	0.9256
Plants	G	0.9556	0.9563	0.9238
	B	0.9478	0.9540	0.9148

**Table 3 entropy-25-00865-t003:** Three-channel correlation analysis of ciphertext images.

Image	Channel	Horizontal	Vertical	Diagonal
	R	0.0002	0.0045	0.0051
C-Pineapple	G	0.0026	0.0012	0.0044
	B	0.0037	0.0046	0.0029
	R	0.0028	0.0049	0.0049
C-Rose	G	0.0055	0.0023	0.0086
	B	0.0034	0.0053	0.0014
	R	0.0010	0.0081	0.0004
C-Plants	G	0.0052	0.0043	0.0025
	B	0.0057	0.0042	0.0033

**Table 4 entropy-25-00865-t004:** Three-channel average NPCR and UACI data.

Image	RGB Average NPCR	RGB Average UACI
C-Pineapple	99.6138%	33.4944%
C-Rose	99.6204%	33.5147%
C-Plants	99.6097%	33.5643%

**Table 5 entropy-25-00865-t005:** Comparison of information entropy of different algorithms.

Algorithm	Average NPCR	Average UACI
Proposed	99.61%	33.42%
Ref. [44]	99.61%	31.60%
Ref. [45]	99.57%	33.38%

**Table 6 entropy-25-00865-t006:** Information entropy data.

Ciphertext Image	R	G	B
Pineapple	7.99925	7.99901	7.99921
Rose	7.99910	7.99930	7.99889
Plants	7.99922	7.99895	7.99912

**Table 7 entropy-25-00865-t007:** Image similarity analysis.

Name	PSNR	SSIM
Pineapple	43.7358	0.982998
Rose	42.9974	0.979976
Plants	43.6438	0.978102
Average	43.4590	0.980358

## Data Availability

The data are contained within the article.

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
