# Peer review of "Quantum Image Encryption Based on Quantum DNA Codec and Pixel-Level Scrambling"

_entropy, 2023, doi:10.3390/e25060865_

Round 1

Reviewer 1 Report

For publication, the paper requires significant revision. Some of my concerns are listed below.

1- the concept of DNA codec requires more explanation and references.

2- The section: "Quantum circuit design" needs a significant revision. The figures and equations need explanation. For example, the circuit of fig. 6 requires a clear explanation of how it relates to rule 1.

3- various equations do not obey quantum principles. For example, Eq. 11 has the notation of a unitary operation, but the operator D_{1,6} defined in Eq. 8 is not unitary. Furthermore, the dimension of state |\psi> and D_{1,6} do not match. In other words, \psi_1, defined in Eq. 10, corresponds to 2n qubits; however, D_{1,6}, defined in Eq. 8, seemingly corresponds to 7 qubits and is independent of dimension n.

4- quantum encryption algorithms, besides safety analysis, requires reversibility analysis which needs to be added to the article. 

Reviewer 2 Report

This manuscript proposed a quantum image encryption scheme based on quantum DNA codec and pixel-level scrambling. Some concerns are listed as follows.

1.     Why the first letter of ‘Quantum DNA Codec’ in the title are capitalized?

2.     Why choose the NCQI model to represent the quantum color digital image?

3.     Line 144, the symbol ‘I’ should be ‘i’.

4.     In Figure 5, what’s the effect of two ancilla qubits. Some explanation should be given in the text.

5.     The authors give the Eq. (8) and (9), but the derivation process should also be given. Why only D1,6 and D1,7 are given? How about D1,2…D1,5 and D1,8?

6.     The ancilla qubits are not appeared in Figure 6.

Reviewer 3 Report

For secure quantum information processing in the future, encryption to quantum images is expected to be one of the promising cryptographic techniques. The authors have introduced efficient way for DNA encoding with quantum Hilbert scrambling for encryption to quantum images. They have found a potential for high security in their scheme, showing that it is more secure than conventional schemes. A quantum version of DNA coding has not been extensively researched; thus, their work provides good insight into the quantum applicability of DNA coding to realizes a large secret key space and strong secret key sensitivity. Overall, this manuscript is clearly written, with nice readability, and the technical content is understandable, clearly showing their scheme. Thus, I recommend the publication of this manuscript in Entropy.

Reviewer 4 Report

The authors analyze the generation of images by using Quantum Principles. The purpose is to analyze and improve the processes of encryption, among other issues. I find the paper interesting but the introduction lacks credibility and I suggest the authors to improve it. For example, the authors said:

"After introducing quantum mechanics in 1982, 25 Feynman went on to create the universal quantum computer[3], a quantum computing 26 device. Deutsch et al.[4] claim that a class of actively solvable problems by quantum computers has been discovered"

First, Feynman did not introduce Quantum Mechanics, he rather understood it in a different way. But Quantum Mechanics exists since 1905, when the concept of photon was introduced. Additionally, I suggest the authors to talk more about certain fundamental issues of Quantum Mechanics, which can help the readers to first enter to the Quantum world and subsequently then get involved into the practical purpose of the article. For example, it would be nice to mention how Quantum Mechanics can fundamentally help in the encryption. Additionally, it would be nice to mention certain weird aspects of Quantum Mechanics, like for example the Hardy's paradox, appearing in the references Phys. Rev. Lett. 68, 2981; Phys. Rev. Lett. 71 (11): 1665. These together with other interesting aspects of quantum mechanics, must be mentioned in the introduction. This is important because it will help the reader to understand why does QM help to improve the image encryption, etc.

In section 5.2, please explain if the correlation function r, is only limited to only neighbor pixels. 

Finally, please extend the conclusions. 

After doing the revision in agreement with these comments, I will happily revise the paper again.

Round 2

Reviewer 1 Report

I reject this paper,

There are still substantial gaps in the new version of the paper, which make the paper hard to grasp and unconvincing. To fill these gaps, in the first review, I asked the authors to explain DNA codec and figures and the proposed quantum circuits in detail. But I received different responses.

Furthermore, I mentioned some fundamental mistakes in equations, but instead of correcting them, the authors simply removed the equation.

Reviewer 2 Report

This paper can be accepted.

Round 3

Reviewer 1 Report

 Quantum DNA coding is the core of this paper, which needs to be detailed very well. The explicit form of the quantum DNA codec operators is the minimum information expected to be given in the paper. Some equations were given in the first version of the paper (Eq. 8 and  Eq. 9 in the first version), which, as explained in my first review, were fundamentally wrong. But the author could not correct the mistake so they remove the equations and tried to give a black-box description to the quantum codec operation.  In short, the paper contains much ambiguity, which hinders the readers from following the paper. Therefore, I can not support the paper for publication; rejection is my final decision.